# Frequency Sweep Keying CDMA for Reducing Ultrasonic Crosstalk

**DOI:** 10.3390/s22124462

**Published:** 2022-06-13

**Authors:** Ga-Rin Park, Sang-Ho Park, Kwang-Ryul Baek

**Affiliations:** 1School of Electric and Electronic Engineering, Pusan National University, Busan 46241, Korea; parkhm7446@pusan.ac.kr (G.-R.P.); propia@pusan.ac.kr (S.-H.P.); 2School of Electronic Engineering, Pusan National University, Busan 46241, Korea

**Keywords:** frequency sweep keying, CDMA ultrasound, ultrasonic ranging system, crosstalk reduction, autonomous driving, front–rear collision avoidance

## Abstract

Various sensors are embedded in automobiles to implement intelligent safety technologies such as autonomous driving and front–rear collision avoidance technology. In particular, ultrasonic sensors have been used in the past because they have an accuracy of centimeters to sub-centimeters in air despite their low cost and low hardware complexity. Recently, the crosstalk problem between ultrasonic sensors has been raised because the number of ultrasonic sensors in the unit space has increased as the number of vehicles increases. Various studies have been conducted to solve the crosstalk, but a demodulation error occurs when signals overlap. Therefore, in this paper, we propose a method that is robust to ultrasonic signal overlap, is robust even at shorter code length, and has reduced time of flight (TOF) error compared to the existing method by applying frequency sweep keying modulation based on code division multiple access (CDMA). As a result of the experiment, the code was detected accurately regardless of the overlap ratio of the two signals, and it was robust even in situations where the power of the two signals was different. In addition, it shows an accurate TOF estimation even if the *ID* code length is shorter than the existing on–off-keying, frequency shift keying, and phase shift keying methods.

## 1. Introduction

Ultrasonic sensors have been widely used in the fields of distance measurements [1,2,3], positioning [4,5,6], non-destructive testing [7,8], and human body inspection [9,10,11]. In particular, they have been used from the past to implement intelligent safety technologies such as autonomous driving of automobiles and front and rear collision avoidance [12,13,14], because of their centimeter to sub-centimeter accuracy in air, low cost, and low hardware complexity. The use of ultrasonic sensors is increasing with the increasing number of vehicles. Owing to the increase in the number of sensors used and sensor arrays for covering a wider range or increasing accuracy, the number of ultrasonic sensors in the unit space has been increased [15,16]. As the result, the problem of crosstalk between ultrasonic sensors has emerged. When a measurement is performed in an environment with a plurality of ultrasonic sensors, an ultrasonic signal transmitted from one sensor other than the sensor to be measured is often received. Owing to such signal interference, the measurement error becomes very large or the system may not operate normally [17].

Several multiple access schemes that guarantee orthogonality between signals have been proposed to solve the signal interference problem. First, time division multiple access (TDMA) is a method in which each node performs TOF measurements at different times to prevent signal interference [18,19,20]. Orthogonality between signals is ensured because the next transmission is performed after receiving a previous transmitted signal. Even after the signal is received by the receiver, the ultrasonic wave remains in the medium and may interfere with the next transmitted signal, so sufficient time is required for the energy of the ultrasonic wave to be sufficiently attenuated. In addition, it is difficult to secure real-time performance using TDMA because the ultrasonic transmission/reception period of each node becomes longer in proportion to the number of sensors in a method that guarantees orthogonality through time. Second, frequency division multiple access (FDMA) is a method in which each node uses a different frequency band by dividing the frequency spectrum [21,22]. A wideband ultrasonic sensor or multiple narrowband ultrasonic sensors with different center frequencies should be used to apply FDMA because a typical ultrasonic sensor has a narrow band. On the other hand, the method of using a plurality of narrowband ultrasonic sensors with different center frequencies increases the complexity of the hardware. Additionally, a wideband ultrasonic sensor is difficult to obtain and is more expensive than a narrowband ultrasonic sensor. Therefore, code division multiple access (CDMA) is widely used to prevent crosstalk among ultrasonic sensors.

CDMA is a method to assign orthogonality between signals by assigning different *ID* codes to each node [23,24,25,26]. In the single-node ultrasound system, the coded ultrasonic method has been used to improve the measurement distance and TOF measurement accuracy because of the improved signal-to-noise ratio (SNR) of the received signal [27,28]. Multiple access (MA) also has the same advantage and is easy to implement in an ultrasound system. As modulation/demodulation methods, on–off keying (OOK) [29], binary phase shift keying (BPSK) [30], quadrature phase shift keying (QPSK) [30], and frequency shift keying (FSK) [31] have been used. Although these methods have the advantage of simple modulation/demodulation, the orthogonality between signals deteriorates when the signals overlap. In the OOK method, when signals overlap, only 1 is always demodulated. In the PSK method, when the signals overlap, the phase has been changed and demodulation cannot be performed. The FSK method cannot accurately estimate the reception time. To solve this problem, the existing method needs to increase the code length to ensure orthogonality between signals.

In most applications that measure the TOF of echo signals, the longer the code length, the longer the signal transmission time, which adversely affects the measurement frequency of the system and the length of the dead zone, which is an unmeasurable area. In this paper, we propose an ultrasonic CDMA method that performs code modulation using a frequency sweep to solve the problem. It includes a code bit modulator through a narrow frequency sweep using a general high-q piezo transducer and a code demodulator through correlation in the spectrogram of a received signal. The CDMA–frequency sweep keying method proposed in this paper has three purposes: it is robust to signal overlap compared to the existing models; it works even with short code length; it is possible to measure the TOF accurately, based on the TOF measurement application.

## 2. Relative Works

### 2.1. CDMA–On–Off Keying

In the most straightforward way, by applying the on–off keying method to the CDMA code, when the code is “1”, $N_{p}$ ultrasonic pulses are transmitted, and when the code is “0”, the ultrasonic waves are not transmitted for the same time $t=N_{p}/f_{c}$, where $N_{p}$ is the number of pulses transmitted per bit, $f_{c}$ is the ultrasonic center frequency, and $t_{p}$ is the transmission time per bit. Although this method is the most convenient to implement using ultrasound and easy-to-process signals, it has several disadvantages. Because the TOF of the first bit of the *ID* code should be measured, the first bit code of the *ID* code should always be composed of 1. This means that there is always a one-bit loss in the code length. In the case of OOK, if the envelope of the received signal is larger than the threshold, it is demodulated to “1”, and if it is smaller, it is demodulated to “0”. In case of the signal overlap, only “1” is demodulated. Related studies include [5,29].

### 2.2. CDMA–Phase Shift Keying

This is a method of transmitting by changing the phase for $N_{P}$ ultrasonic transmission pulses according to the *ID* code. There are binary phase shift keying for keying 1-bit codes with phases at 180° intervals, and quadrature phase shift keying for keying 2-bit codes with phases at intervals of 90°. It is widely used in narrowband ultrasonic transducers because of their high energy efficiency. On the other hand, when the signals overlap, the phase of the overlapped signals is changed according to the magnitude and phase of the individual signals, which causes an error in demodulation with a different code during demodulation. Related studies include [24,32,33].

### 2.3. CDMA–Frequency Shift Keying

This is a transmission method by changing the frequency of $N_{p}$ ultrasonic transmission pulses based on the center frequency $f_{c}$ according to the *ID* code. In the case of broadband ultrasonic transducers, energy efficiency and modulation/demodulation performance are excellent. In the case of a narrowband ultrasonic transducer, however, the energy efficiency is not good because it transmits using a frequency deviating from the center frequency $f_{c}$, and SNR degradation and sensing range are reduced. No interference between code “1” and code “0” occurs during signal overlap. Related studies include [34,35].

## 3. Proposed Frequency Sweep Keying CDMA

### 3.1. Proposed Frequency Sweep Keying Modulation

The keying modulation method proposed in this paper was implemented using a frequency sweep. As shown in Figure 1, the frequency rising sweep was defined as “1” and the frequency falling sweep was defined as “0”. Equation (1) shows the frequency $f_{k}\left(t\right)$ according to the frequency sweep time and Equation (3) shows the transmitted ultrasonic signal $s_{tx}\left(t\right)$ when transmitted to an ideal transmitter without a bandwidth limitation.
(1)$$f_{k}\left(t\right)=\frac{\Delta f}{\Delta t}k\left(\left(t\;mod\;\Delta t\right)-\frac{\Delta t}{2}\right)+f_{c}$$
(2)$$s_{n,k}\left(t\right)=\sin\left(2\pi f_{k}\left(t\right)\left(t-n\Delta t\right)\right)\left\{u\left(t-n\Delta t\right)-u\left(t-\left(n+1\right)\Delta t\right)\right\}$$
(3)$$s_{tx}\left(t\right)=\overset{L_{c}}{\underset{n=1}{\sum }}s_{n,k}\left(t\right)$$
where k is bit code information and has a value of +1 or −1. When bit code is 1, then k has +1, and k has −1 when bit code is 0. $f_{c}$ is the sweep center frequency and was set to the center frequency of the ultrasonic sensor. The frequency sweep was performed in the upper limit $f_{c}+\Delta f/2$ and lower limit $f_{c}-\Delta f/2$. Δf is the frequency sweep range. Δt is the time to perform the frequency sweep. $s_{n,k}\left(t\right)$ represents the signal corresponding to the nth bit in the code, and the transmitted ultrasound signal $s_{tx}\left(t\right)$ using the ideal transmitter is represented by the sum of the signals corresponding to each bit. Figure 1 shows the transmission signal $s_{tx}\left(t\right)$ on the spectrogram.

### 3.2. Bandwidth Limitation of Ultrasonic Transducers

The ultrasonic transducer consists of a piezoelectric material and a matching layer to improve energy transfer, a backing material to reduce the ringing of the piezoelectric material, and an electrically insulated housing [36]. When an AC signal is applied to an ultrasonic transducer, the piezoelectric material vibrates at the corresponding frequency to generate ultrasonic waves. The amplitude is greatly amplified if the wavelength corresponding to this frequency is equal to half the thickness of the piezoelectric material. This frequency is called the fundamental resonant frequency. Most inexpensive ultrasonic transducers exhibit high-Q characteristics with a narrow bandwidth with this resonant frequency as the center frequency. Owing to this resonance characteristic of the ultrasonic transducer, ultrasonic waves with sufficient energy to measure the distance are not transmitted when only one pulse signal is applied. Several pulse signals in the form of a burst must be continuously applied to transmit an ultrasonic signal of sufficient energy to detect the ultrasonic wave reflected back from the measurement target; several pulse signals in the form of a burst must be continuously applied. $N_{p}$, which is the number of pulses transmitted per bit, is appropriately selected in consideration of this.

The signal power in the center frequency band is strongest when a frequency sweep is performed because of the bandwidth limitation of the ultrasonic transducer, as shown in Figure 2, and sharp signal attenuation occurs outside the bandwidth. Therefore, if the frequency sweep range, Δf, is set too wide, the ratio of the time of the area where the signal size to the time of the one-bit signal is small increases, which is not good. Therefore, considering the bandwidth of the ultrasonic transducer, the frequency sweep range should be selected within the frequency $f_{-20\mathrm{dB}}$ section in which the signal amplitude is −20 dB compared to the center frequency.
(4)$$f<\frac{\left|f_{c}-f_{-20\mathrm{dB}}\right|}{2}$$

Figure 3 presents time–frequency spectrograms of a one-bit transmission signal considering the bandwidth-limiting effect of an ultrasonic transducer.

There is only one resonant peak because we are targeting a typical, easy-to-use narrowband transducer. However, in the case of a wideband transducer, harmonic frequencies may be included, so the frequency sweep range Δf must be carefully selected.

### 3.3. Proposed Frequency Sweep Keying Demodulation

The ultrasonic waves propagating into the air through the preceding processes are reflected from the measurement target and received by the ultrasonic receiver. The received ultrasonic signal is defined as $s_{rx}\left(t\right)$, and the spectrogram [37] of the received signal can be seen (Figure 4). A cross-correlation is performed for code separation of the received ultrasound signal. The received ultrasound signal is converted to a signal on the time–frequency spectrogram for simultaneous analysis in both the time and frequency domains. Equations (5)–(7) are used to convert to a spectrogram as follows.
(5)$$S_{rx}\left(m,f\right)=STFT\left\{s_{rx}\left[n\right]\right\}\left(m,f\right)=\int_{-\infty }^{\infty }s_{rx}\left[n\right]w\left[n-m\right]e^{-j2\pi fn}dt$$
(6)$$w\left[n\right]=L\cdot w_{0}\left(\frac{L}{N}\left(n-\frac{N}{2}\right)\right)=\frac{I_{0}\left[\beta \sqrt{1-{\left(\frac{2n}{N}-1\right)}^{2}}\right]}{I_{0}\beta },\;0\leq n\leq N\;$$
(7)$$spectrogram\left\{s_{rx}\left[n\right]\right\}\left(m,f\right)={\left|S_{rx}\left(m,f\right)\right|}^{2}$$
where $S_{rx}\left(m,f\right)$ is a result of performing a short-time Fourier transform on the received ultrasound signal $s_{rx}\left(t\right)$. The Kaiser window $w\left[n\right]$ is used, where $I_{0}$ is the zero-order modified Bessel function, L is window duration, N is window length −1, and β is a coefficient indicating the characteristic of the Kaiser window. The spectrogram is obtained by squaring the magnitude of $S_{rx}\left(m,f\right)$.

Demodulation of the received ultrasound signal is performed through a cross-correlation with a reference bit signal corresponding to one bit having values of “1” and “0”. Thereafter, CDMA decoding is performed through cross-correlation between the demodulated signal and the *ID* code. The reference bit signals ${\hat{s}}_{1}$ and ${\hat{s}}_{0}$ having values of “1” and “0”, respectively, are the estimated signals corresponding to one bit considering the bit value, information on the frequency sweep, and characteristics of the transmitting/receiving ultrasonic transducer.
$${\hat{s}}_{1}=s_{0,1}\left[n\right]{*}_{BPF}\left[n\right]\;$$
(8)$${\hat{s}}_{0}=s_{0,0}\left[n\right]{*}_{BPF}\left[n\right]$$
where $h_{BPF}\left[n\right]$ is a time domain function obtained by performing an inverse Laplace transform on the transfer function $H_{BPF}\left(s\right)$ of the high-Q bandpass filter of the ultrasonic transducer. If $H_{BPF}\left(s\right)$ is measured using the frequency characteristic of the ultrasonic transducer, it is more precise, but it can also be used as the equation of the second-order bandpass filter.
(9)$$H_{BPF}\left(s\right)=\frac{s}{s^{2}+\left(\frac{2\pi f_{c}}{Q}\right)s+{\left(2\pi f_{c}\right)}^{2}}$$
where *Q* is $2f_{c}/\left|f_{c}-f_{-3\mathrm{dB}}\right|$, which means the *Q* value of the filter. The reference signals ${\hat{s}}_{1}$ and ${\hat{s}}_{0}$ are then converted into signals ${\hat{S}}_{1}\left(m,\;f\right)$ and ${\hat{S}}_{0}\left(m,\;f\right)$ on the spectrogram using Equations (1)–(3). From ${\hat{S}}_{1}\left(m,\;f\right)$ and ${\hat{S}}_{0}\left(m,\;f\right)$, the part corresponding to the frequency and time domain of interest is composed of the reference cross-correlation masks $M_{1}\left(m,f\right)$ and $M_{0}\left(m,f\right)$. The frequency domain of interest corresponds to the sweep frequency range $\left(f_{c}+\frac{\Delta f}{2},\;f_{c}-\frac{\Delta f}{2}\right)$ and the time domain of interest corresponds to the transmission time of one bit $\left(-\Delta t/2,\;\Delta t/2\right)$

Finally, cross-correlations between the received ultrasound spectrogram signal $S_{rx}\left(m,f\right)$ and the reference bit masks $M_{1}\left(m,f\right)$ and $M_{0}\left(m,f\right)$ are obtained. When cross-correlating, the frequency domain limits the range to the frequency domain of interest and performs a time axis shift.
$$\rho_{1}\left[n\right]=\overset{\Delta n/2}{\underset{m=-\Delta t/2}{\sum }}\overset{f_{c}+\Delta f/2}{\underset{f=f_{c}-\Delta f/2}{\sum }}S_{rx}\left(n+m,f\right)M_{1}\left(m,f\right)$$
(10)$$\rho_{0}\left[n\right]=\overset{\Delta n/2}{\underset{m=-\Delta t/2}{\sum }}\overset{f_{c}+\Delta f/2}{\underset{f=f_{c}-\Delta f/2}{\sum }}S_{rx}\left(n+m,f\right)M_{0}\left(m,f\right)$$

If the cross-correlation value is greater than or equal to the threshold value $\rho_{th}$, it is determined that the received ultrasound signal has a value of “1” and “0”, respectively, and the received ultrasound signal is decoded as the reception code $c_{rx,1}\left[n\right]$ and $c_{rx,0}\left[n\right]$. Because the result value of the correlation operation is proportional to the power of the received signal, the threshold value should also be set in proportion to it. Therefore, the threshold value is determined based on the cross-correlation value between the received signal and the *X* mask. The *X* mask is calculated as the maximum value of the reference bit “1” mask and “0” mask, as expressed in the following equation.
(11)$$M_{X}\left(m,f\right)=\max\left(M_{1}\left(m,f\right),\;M_{0}\left(m,f\right)\right)$$
(12)$$\rho_{X}\left[n\right]=\overset{\Delta n/2}{\underset{m=-\Delta t/2}{\sum }}\overset{f_{c}+\Delta f/2}{\underset{f=f_{c}-\Delta f/2}{\sum }}S_{rx}\left(n+m,f\right)M_{X}\left(m,f\right)$$

A reference bit cross-correlation value of a code matching a received signal and a cross-correlation value of an *X* mask have similar values. The reference bit cross-correlation value with the incorrect code has a smaller value to the cross-correlation value of the *X* mask. Therefore, the value obtained by multiplying the cross-correlation value $\rho_{X}$ of the *X* mask by the threshold coefficient $\gamma_{th}$ is determined as the threshold value. The threshold coefficient $\gamma_{th}$ has a value greater than 0 and less than or equal to 1.
$$c_{rx,1}\left[n\right]=\left\{\begin{array}{l} 1,\;\;\rho_{1}\left[n\right]>\gamma_{th}\rho_{th}\left[n\right]\\ 0,\;\;\;\;\;\;\;\;\;\;otherwise \end{array}\right.$$
(13)$$c_{rx,0}\left[n\right]=\left\{\begin{array}{l} 1,\;\;\rho_{0}\left[n\right]>\gamma_{th}\rho_{th}\left[n\right]\\ 0,\;\;\;\;\;\;\;\;\;\;otherwise \end{array}\right.$$

In the mixed ultrasound signal, reference bits “1” and “0” can be received simultaneously, so the reception code is a state in which no signal is received $c_{rx,1}$ = 0, $c_{rx,0}$ = 0), no bit “1” received ($c_{rx,1}$ = 1, $c_{rx,0}$ = 0), no bit “0” received ($c_{rx,1}$ = 0, $c_{rx,0}$ = 1), and bit “1” and bit “0” are received simultaneously ($c_{rx,1}$ = 1, $c_{rx,0}$ = 1). It is divided into four cases.

### 3.4. CDMA Decoding

The dimensions of two codes are matched to perform a correlation between the demodulated codes $c_{rx,1}\left[n\right]$, $c_{rx,0}\left[n\right]\;$ and *ID* code $C_{ID.}$ After dividing by bit “1” and bit “0”, $c_{ID,1}$ and $c_{ID,0}$ are calculated by upsampling, as shown in the following equation.
$$c_{ID,1}\left[n\right]=C_{ID}{\left[m\right]}_{↑L}$$
(14)$$c_{ID,0}\left[n\right]={\left(1-C_{ID}\left[m\right]\right)}_{↑L}$$
where L is the upsampling factor and was calculated as $\Delta t/f_{S}$. The correlation between the reception codes $c_{rx,1}\left[n\right]$, $c_{rx,0}\left[n\right]$ (*t*) and the *ID* codes $c_{ID,1}$ and $c_{ID,0}$ was carried out.
$$\rho_{c1}\left[n\right]=\overset{L_{u}L_{ID}}{\underset{m=0}{\sum }}c_{rx,1}\left[n+m\right]c_{ID,1}\left[m\right]$$
(15)$$\rho_{c0}\left[n\right]=\overset{L_{u}L_{ID}}{\underset{m=0}{\sum }}c_{rx,0}\left[n+m\right]c_{ID,0}\left[m\right]$$
(16)$$\rho_{c}\left[n\right]=\rho_{c1}\left[n\right]+\rho_{c0}\left[n\right]$$
where, if $\rho_{c}\left[n\right]$ is the same as $L_{ID}$, the corresponding *ID* is decoded. Hence, the time when $\rho_{c}\left[n\right]=L_{ID}$ is calculated as TOF $t_{f}$, as shown in the following equation.
(17)$$n_{f}=min\left\{x|\rho_{c}\left[x\right]=L_{ID}\right\}$$
(18)$$t_{f}=n_{f}/f_{S}$$

## 4. Experiment

### 4.1. Experiment Environment

As shown in Figure 5, the ultrasonic transceiver system consists of personal computer (PC), microcontroller unit (MCU), amplifier (AMP), driver, and ultrasonic transducer. The ultrasonic transducer has one receiver (Rx) and several transmitters (Tx) to configure the environment where crosstalk occurs. The driver amplifies the power of the MCU’s transmit signal to increase the strength of the ultrasonic transmit signal. The amplifier amplifies the voltage of the signal of the ultrasonic receiver and transmits it to the analog digital converter (ADC) of MCU. The signal-to-noise ratio (SNR) of the signal is improved by using a low-noise amplifier. The MCU receives the trigger command from the PC, generates a transmit signal, and transfers the received signal to the PC. It can control the frequency in a burst form so that the transmission signal can be modulated with the signal of the proposed method and the existing method. It can also adjust the time delay so that the point of crosstalk can be changed. The PC demodulates the received signal and executes the CDMA algorithm.

Figure 6a shows the ultrasonic transceiver used in the experiment. The ultrasonic sensor uses Hagisonic’s HG-L40DC, has a bandwidth of 4 kHz, and a directivity angle of 65° at a center frequency of 40 kHz. In addition, the effect of multipath fading and reverberation was minimized by attaching an ultrasonic sound-absorbing agent around the ultrasonic sensor to absorb ultrasonic waves other than the line of sight. As shown in Figure 6b, an ultrasonic signal interference environment was constructed by configuring multiple transmitters in one receiver in an indoor environment 10 m in width and 10 m in height. Table 1 lists the parameter values used in the experiment.

### 4.2. One-Bit Signal Overlap Experiment

An experiment was performed to verify whether the frequency sweep keying proposed in this paper can demodulate the individual signals when signals overlap. When two signals of multi-code overlap, the overlap ratio and signal power of each bit are the same, so an overlap experiment was performed on a 1-bit signal. In case of multi-code, overlap occurs in both directions for one bit, so at least 50% overlap occurs. Figure 7 and Figure 8 show demodulation results when two transmitters transmit the same bit codes, “1” and “1”, and when two transmitters transmit different bit codes, “1” and “0”. The top row represents original received signals. The second row represents signals as spectrograms. The bottom row represents cross-correlation results between second-row signals and *X* mask. In terms of columns, the left column shows when signals overlap by 0%, the middle column shows when signals overlap by 50%, and the right column shows when signals overlap by 100%. Even with the same overlap ratio, when signals overlap, attenuation and constructive interference occur depending on the phase of each signal, and accordingly, the shape of the received waveform and spectrogram changes slightly. In correlation calculation, the threshold value is important because the correlation value varies depending on the scale. As a result of the experiment, each code is accurately detected even at all overlap ratios. In particular, even if other bit codes overlap 100%, both code 0 and code 1 are detected.

A code with a small power length is difficult to demodulate when two signals with a large difference in signal power are mixed. Therefore, an experiment was conducted to determine the code detection ratio according to the difference between the two signal powers. The distance of the transmitter was adjusted so that the ratio of the code 1 signal voltage to the code 0 signal voltage was 1.0, 0.8, 0.6, 0.4, and 0.2, and the overlap was randomized by adjusting the time delay. In the case of code 0, the signal power was strong and it was always received. In the case of code 1, which has a weak signal power, as shown in Figure 9, the code detection ratio rapidly decreased when the signal power ratio was lowered to less than 0.4. When the signal power ratio was 0.2, the code detection ratio decreased to approximately 10% at an overlap of 40% or more. In this case, it was impossible to detect a signal code in a multi-code because the detection ratio is low at 50% or more of overlap. In conclusion, in the proposed frequency sweep keying, the demodulator operates normally at a signal power ratio of 0.4 or higher.

### 4.3. TOF Error Comparison with Other Demodulators

An experiment was conducted to verify that the proposed algorithm is robust against crosstalk through comparative verification with OOK, PSK, and FSK introduced in Relative Works. First, an experiment was conducted to verify that the proposed method can apply a shorter code length than the existing method. As a verification method, TOF accuracy according to code lengths 2, 4, 8, 16, 32, 64, and 128 was compared. We placed the two transmitters at the same distance and the receiver received a mixed signal of the two code signals. Table 2 shows the comparison result of TOF accuracy according to *ID* code length between the proposed method and the existing method. The TOF errors were compared through a root mean square (RMS) operation. For each *ID* length, the overlap was measured 100 times in 10% steps from 0% to 100%, for 1100 times. In the CDMA process performed after demodulation, the TOF estimated the time point with the greatest correlation as the reception time point. Therefore, when a code error occurs in the demodulator, the maximum correlation value is lowered, and the error of the estimated position becomes very large. Therefore, the *ID* code demodulation has failed if the TOF error is greater than ∆t/2, 1 ms, which is half of one bit. As a result of the experiment, among the existing methods, OOK and FSK showed a tendency to decrease the TOF error as the code length increased.

In OOK, PSK, and FSK, the demodulation error occurred because the error was 1 ms or more in *ID* length 16 or less. That is, the *ID* detection failed and the TOF error occurred larger than 1 ms. In particular, the TOF error was significantly larger even when the code length was increased in PSK. In contrast, the proposed method had low demodulation error and high TOF accuracy regardless of *ID* length. Therefore, in the case of the proposed method, accurate TOF can be estimated even if two transmitters use only a minimum code length of 2 in an environment where there is a possibility of collision.

Second, an experiment was conducted to verify that the proposed method is more robust than the existing method even in the signal overlap situation. In the same experimental environment as the first experiment, the overlap ratio of the mixed signal was adjusted by changing the transmission time point to the two transmitters. If the TOF error was less than 1 ms, it was judged that the code detection was successful. Figure 10 shows the experimental results. The detection ratio was calculated by measuring each 100 times in 10% units from 0% overlap to 100% overlap. As a result of the experiment, the proposed method showed a detection ratio of 0.93 or more regardless of the overlap ratio. However, in the case of the existing method, the detection ratio tended to decrease as the overlap ratio increased. In particular, in the case of PSK, the decrease was even more pronounced.

## 5. Conclusions

In this paper, frequency sweep keying–CDMA was proposed to solve the crosstalk problem among ultrasonic sensors that occurs as the density of ultrasonic sensors increases. The method proposed in this paper included a code bit modulator through a narrow frequency sweep using a general high-queue piezo transducer and a demodulator through the correlation in the spectrogram of the received signal. Due to the characteristics of the demodulation method, even if signals are mixed, the code and reception time of each signal can be accurately detected.

To verify the algorithm proposed in this paper, the experiment was conducted by configuring an experiment environment. As a result of the one-bit overlap experiment, the frequency sweep keying–CDMA proposed in this paper was verified to correctly demodulate the code regardless of the signal overlap ratio. In addition, demodulation was possible when the signal intensity ratio of the mixed signal was 1:0.4 or more. As a result of comparison verification experiments with the existing OOK, PSK, and FSK methods, the existing methods showed a large TOF error at code lengths of 16 or less, but the proposed method accurately estimated TOF from code length 2. In addition, the existing methods tended to lower the code detection ratio as the overlap ratio increased, but the proposed method showed a high detection ratio of at least 0.93 regardless of the overlap ratio.

Experimental results may vary depending on various experimental environmental factors such as ultrasound equipment, environment, and SNR, but it is expected to have a similar tendency to the experimental results in this paper. Compared to the existing method, the algorithm proposed in this paper can demodulate the code more accurately in the mixed signal, so it is efficient when used in an environment with a large number of ultrasound devices.

On the other hand, there are some limitations. First, in the case of a moving object, the Doppler effect, that is, a frequency shift phenomenon, occurs due to the speed difference. The greater the speed difference, the greater the frequency shift. The proposed demodulator performs code detection by correlating the code mask based on the center frequency. If a slight frequency shift occurs, demodulation will be performed normally. When a large frequency shift occurs, the correlation process is not normally performed because the spectrogram of the received signal deviates a lot from the center frequency. Therefore, there is a problem that the code cannot be detected. Second, when a heterogeneous signal other than the signal of the proposed method is received, a demodulation error may occur. The demodulator proposed in this paper sets the value obtained by multiplying the correlation value of the *X* mask by a coefficient as a threshold value. It is determined whether 1 is received, 0 is received, or simultaneously received according to whether each correlation value of 1 mask and *X* mask, and 0 mask and *X* mask is greater or less than the threshold. Even if an arbitrary signal is received, if the correlation value of the 1 mask and 0 mask is less than the threshold, code 0 or 1 is treated as not being received. OOK and PSK are detected differently depending on the threshold coefficient value because they have most of the energy at the center frequency. If the threshold coefficient is increased, OOK and PSK are not detected, but the probability that the method proposed in this paper will not be detected also increases. In the case of FSK, since energy is located at a frequency shifted from the center frequency, it is hardly detected.

Research that can solve the limitations mentioned as further works should be conducted. First, a study on the case where the Doppler effect occurs should be conducted. When the Doppler effect occurs, the frequency of the signal shifts up or down in the spectrogram. As a simple method, there is a method of performing correlation not only on the center frequency but also on other frequency ranges. However, the real-time performance of this method is poor because the amount of computation is very large. Therefore, if research such as estimating how much frequency shift occurs in the spectrogram is progressed, it is expected that the performance will be improved even in the situation where the Doppler effect occurs. Second, the processing of heterogeneous signals, not the signals proposed in this paper, should proceed. The shape on the spectrogram is clearly different according to each modulation method. Therefore, if a learning algorithm that can learn the characteristics of two-dimensional data, such as CNN, is used, only the signal proposed in this paper can be processed by classifying which modulation signal is received. Then, it is expected that it will be much stronger, even in the outdoor environment where arbitrary ultrasonic signals can enter.

## Figures and Tables

**Figure 1 sensors-22-04462-f001:**
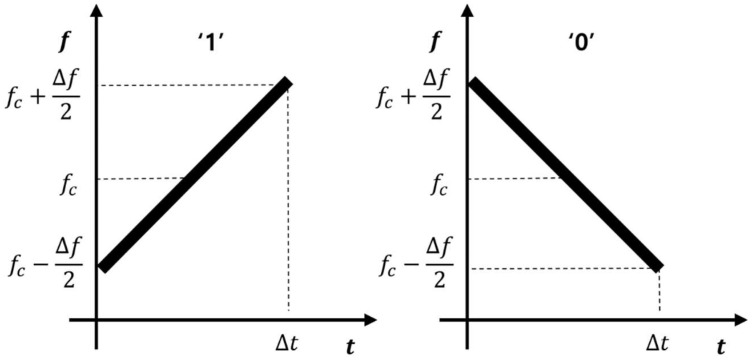
Frequency sweep keying.

**Figure 2 sensors-22-04462-f002:**
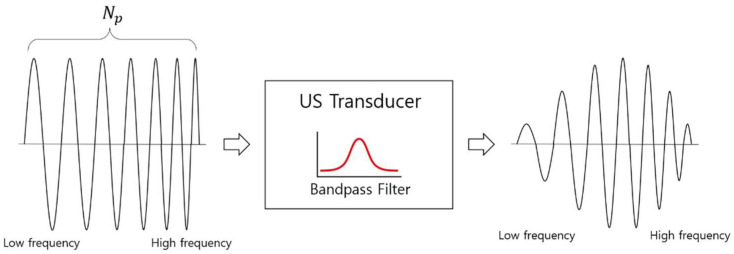
Transmission signal change due to ultrasonic band limiting.

**Figure 3 sensors-22-04462-f003:**
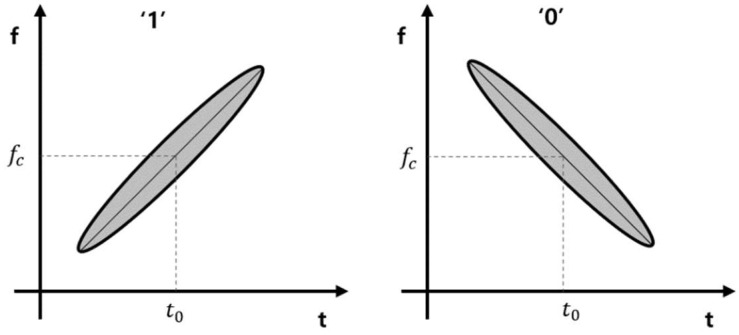
Band-limited bit signal on the spectrogram.

**Figure 4 sensors-22-04462-f004:**
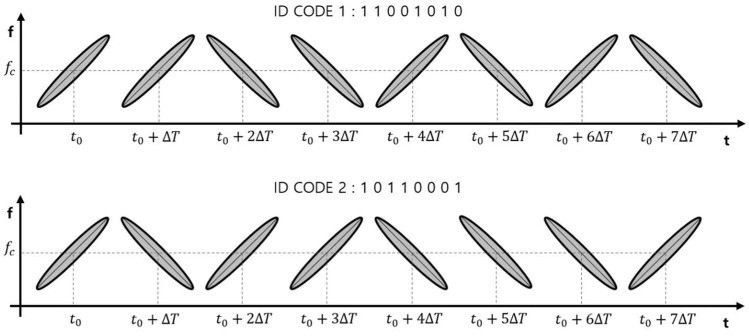
Example of the received signal according to the code on the spectrogram.

**Figure 5 sensors-22-04462-f005:**
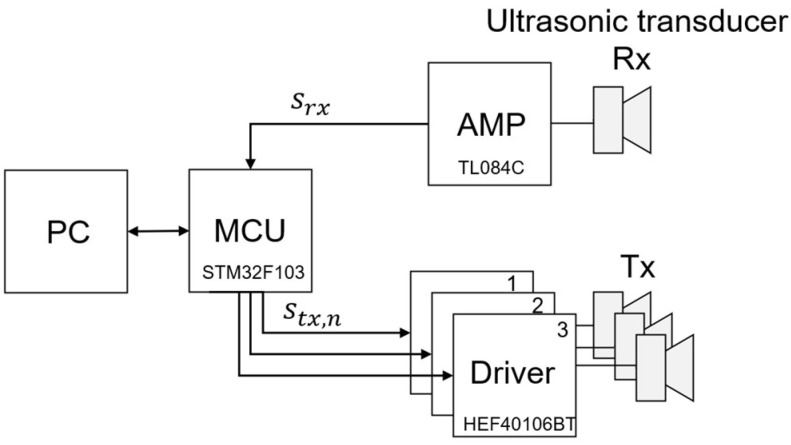
Ultrasonic transceiver system block diagram for proposed algorithm verification.

**Figure 6 sensors-22-04462-f006:**
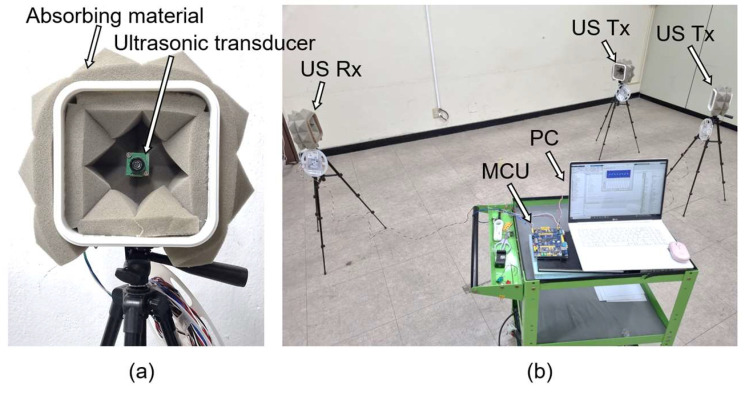
Ultrasonic transmission/reception test environment: (**a**) ultrasonic transducer, (**b**) experiment environment.

**Figure 7 sensors-22-04462-f007:**
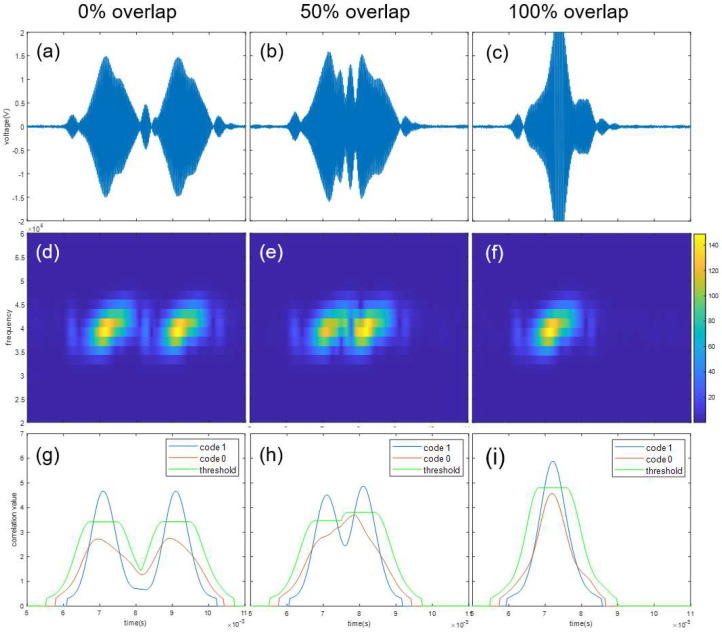
Demodulation results when two transmitters transmit the same bit codes, “1” and “1”. Shown are (**a**,**d**,**g**)—0% overlap; (**b**,**e**,**h**)—50% overlap; (**c**,**f**,**i**)—100% overlap; (**a**–**c**)—received signal; (**d**–**f**)—spectrogram; and (**g**–**i**)—code mask correlation result and threshold value.

**Figure 8 sensors-22-04462-f008:**
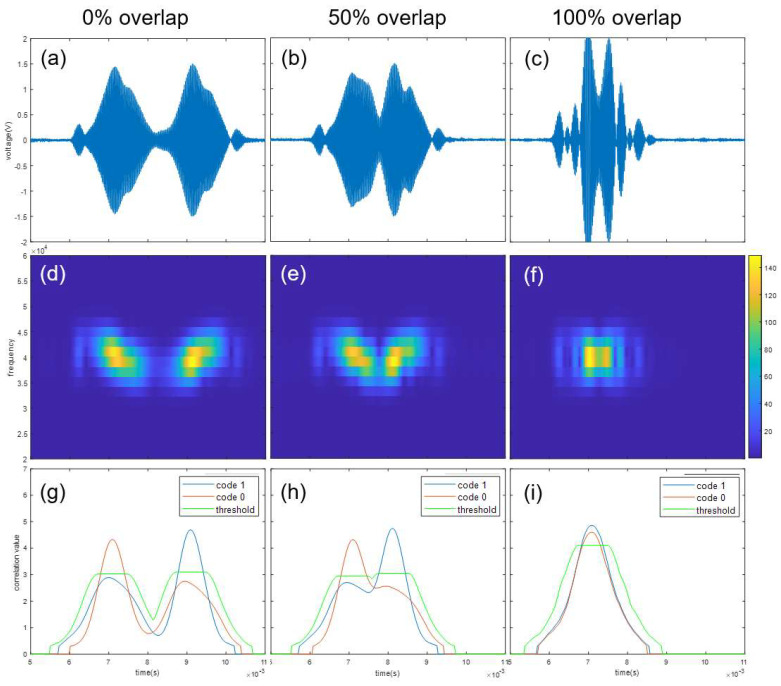
Demodulation results when two transmitters transmit the same bit codes, “1” and “0”. Shown are (**a**,**d**,**g**)—0% overlap; (**b**,**e**,**h**)—50% overlap; (**c**,**f**,**i**)—100% overlap; (**a**–**c**)—received signal; (**d**–**f**)—spectrogram; and (**g**–**i**)—code mask correlation result and threshold value.

**Figure 9 sensors-22-04462-f009:**
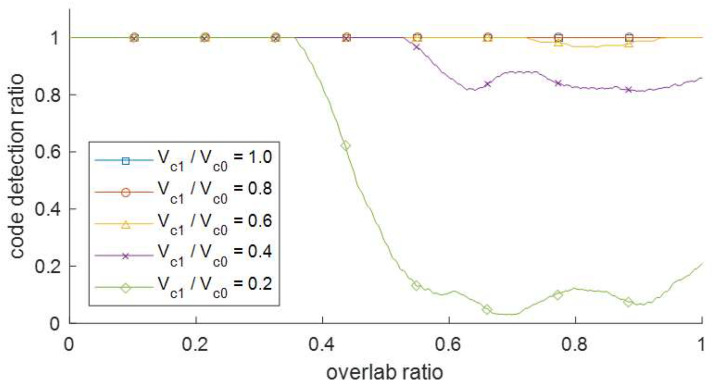
Code detection results of a signal with low signal power when two transmitters transmit with different signal power.

**Figure 10 sensors-22-04462-f010:**
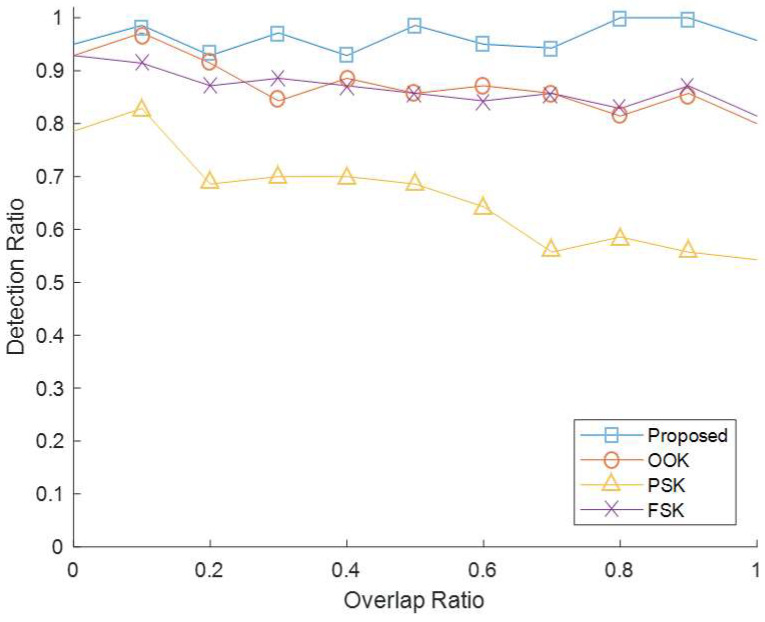
Comparison of detection ratio according to overlap of the proposed method and the existing method.

**Table 1 sensors-22-04462-t001:** Experimental parameters.

Symbol	Parameter	Value	Unit
$f_{c}$	Ultrasonic center frequency	40	kHz
	Transducer frequency bandwidth	4	kHz
$f_{s}$	ADC sampling rate	1	MHz
Δf	Sweep frequency range	8	kHz
Δt	Sweep time	2	ms
β	Keiser window coefficient	18	
$L_{w}$	Keiser window length	512	
$\gamma_{th}$	Threshold coefficient	0.7	

**Table 2 sensors-22-04462-t002:** Comparison of RMS TOF error according to *ID* length of the proposed method and the existing method.

Code Length	RMS and Standard Deviation of TOF Error (ms)			
	Proposed	OOK	PSK	FSK
2	0.257 ± 0.126	66.992 ± 29.414	68.831 ± 30.126	66.638 ± 29.207
4	0.516 ± 0.258	32.187 ± 15.625	33.244 ± 16.149	32.339 ± 15.629
8	0.362 ± 0.180	3.138 ± 1.548	5.671 ± 2.756	4.235 ± 2.061
16	0.143 ± 0.070	2.562 ± 1.285	10.019 ± 4.692	4.288 ± 2.134
32	0.088 ± 0.021	0.433 ± 0.024	18.063 ± 8.321	1.965 ± 0.986
64	0.330 ± 0.025	0.445 ± 0.019	26.101 ± 11.954	0.298 ± 0.056
128	0.390 ± 0.027	0.449 ± 0.018	13.129 ± 5.844	0.204 ± 0.050

## Data Availability

Not applicable.

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
