# Peer review of "Frequency Sweep Keying CDMA for Reducing Ultrasonic Crosstalk"

_sensors, 2022, doi:10.3390/s22124462_

Round 1
Reviewer 1 Report
This article needs a complete revision in terms of writing and grammar. Please check the whole article and edit as much as you can. As an example:
--There are several location which the sentences started with "And", this is not a proper way of wording
--Page 2, Section 1, Line 82: change to "This paper is organized as follows:..."
--Page 4, Section 3, Line 154: "must be continuously applied continuously...", repeated word, please revise
--Page 7, Section 3, Line 217: "...the signal power of the received signal..", change to "...the power of the received signal..."
And several other placed which needs revising
+In last paragraph of section 3.1 after the formulas 1-3, also define what is Delta f
+In same paragraph it is mentioned: "Figure 1 shows the transmission..." Figure 1 is something else, please check and revise
+Section 4: The type and model of the sensor mentioned, is it for just the transmitter or for both transmitter and receiver? Please identify both T&R types of sensors
+You must expand the discussion of the paper by explaining the details of remaining challenges and suggestion for future research
Author Response
--There are several location which the sentences started with "And", this is not a proper way of wording
-> We revised all the sentences started with "And".
--Page 2, Section 1, Line 82: change to "This paper is organized as follows:..."
-> This paragraph was eliminated because other reviewer wanted it.
--Page 4, Section 3, Line 154: "must be continuously applied continuously...", repeated word, please revise
-> We revised it.
--Page 7, Section 3, Line 217: "...the signal power of the received signal..", change to "...the power of the received signal..."
-> We changed it.
And several other placed which needs revising
+In last paragraph of section 3.1 after the formulas 1-3, also define what is Delta f
-> We added it as "delta f is the frequency sweep range"
+In same paragraph it is mentioned: "Figure 1 shows the transmission..." Figure 1 is something else, please check and revise
->
As a result of checking, Figure 1 shows the transmission signal on the spectrogram. In the case of an ideal transmitter, on the spectrogram, the transmission signal rises or falls by the frequency modulation width delta f based on the center frequency fc during delta t time.
+Section 4: The type and model of the sensor mentioned, is it for just the transmitter or for both transmitter and receiver? Please identify both T&R types of sensors
->
There are internally generated (M) and externally generated (L) models for this model, and the externally generated (L) model was used. Also, there are transmitter (T), receiver (R), and transceiver (D) models, and the transmitter (D) model was used.
+You must expand the discussion of the paper by explaining the details of remaining challenges and suggestion for future research
->
This has been added to the conclusion.
There are some limitations. First, in the case of a moving object, the Doppler effect, that is, a frequency shift phenomenon, occurs due to the speed difference. The greater the speed difference, the greater the frequency shift. The proposed demodulator performs code detection by correlating the code mask based on the center frequency. If a slight frequency shift occurs, demodulation will be performed normally. When a large frequency shift occurs, the correlation process is not normally performed because the spectrogram of the received signal deviates a lot from the center frequency. Therefore, there is a problem that the code cannot be detected. Second, when a heterogeneous signal other than the signal of the proposed method is received, a demodulation error may occur. The demodulator proposed in this paper sets the value obtained by multiplying the correlation value of the X mask by a coefficient as a threshold value. It is determined whether 1 is received, 0 is received, or simultaneously received according to whether each correlation value of 1 mask and X mask, and 0 mask and X mask is greater or less than the threshold. Even if an arbitrary signal is received, if the correlation value of the 1 mask and 0 mask is less than the threshold, code 0 or 1 is treated as not being received. OOK and PSK are detected differently depending on the threshold coefficient value because they have most of the energy at the center frequency. If the threshold coefficient is increased, OOK and PSK are not detected, but the probability that the method proposed in this paper will not be detected also increases. In the case of FSK, since energy is located at a frequency shifted from the center frequency, it is hardly detected.
Research that can solve the limitations mentioned as Further works should be conducted. First, a study on the case where the Doppler effect occurs should be conducted. When the Doppler effect occurs, the frequency of the signal shifts up or down in the spectrogram. As a simple method, there is a method of performing correlation not only on the center frequency but also on other frequency ranges. However, the real-time performance of this method is poor because the amount of computation is very large. Therefore, if research such as estimating how much frequency shift occurs in the spectrogram is progressed, it is expected that the performance will be improved even in the situation where the Doppler effect occurs. Second, the processing of heterogeneous signals, not the signals proposed in this paper, should proceed. The shape on the spectrogram is clearly different according to each modulation method. Therefore, if a learning algorithm that can learn the characteristics of two-dimensional data such as CNN is used, only the signal proposed in this paper can be processed by classifying which modulation signal is received. Then, it is expected that it will be much stronger even in the outdoor environment where arbitrary ultrasonic signals can enter.
Reviewer 2 Report
The manuscript introduces a new method of signal modulation and demodulation for signal overlap handling. A frequency sweep keying modulation and corresponding demodulation is proposed with theoretical study and experimentally validation. It is proved that with the proposed modulation method, the time of flight estimation is substantially better than other common modulation methods (e.g., OOK, PSK, and FSK). Moreover, different factors are evaluated for the method such as the signal power and overlap ratio. The overall concept of the paper is well-established, and the evaluation method is concrete. Some minor comments can be considered before the publication on Sensors.
Comments:
1. Some acronyms are used before indicating the full name, for example, TOF and CDMA in the abstract.
2. The study did not show the robustness of the proposed method to different types of signal noise. It would be better to articulate the possible performance when dealing with the noisy signals as the frequency sweeping may expand certain noise in the signal.
3. For some piezo-electric transducers, there will be multiple resonance peaks in a wide range of frequencies, specifically at the harmonic frequency of the “center frequency”. Thus, the transmission curve may include multiple peaks, and a careful bandwidth selection should be made to define the range of f-20Db.
Author Response
The manuscript introduces a new method of signal modulation and demodulation for signal overlap handling. A frequency sweep keying modulation and corresponding demodulation is proposed with theoretical study and experimentally validation. It is proved that with the proposed modulation method, the time of flight estimation is substantially better than other common modulation methods (e.g., OOK, PSK, and FSK). Moreover, different factors are evaluated for the method such as the signal power and overlap ratio. The overall concept of the paper is well-established, and the evaluation method is concrete. Some minor comments can be considered before the publication on Sensors.
Comments:
- Some acronyms are used before indicating the full name, for example, TOF and CDMA in the abstract.
-> We revised it.
- The study did not show the robustness of the proposed method to different types of signal noise. It would be better to articulate the possible performance when dealing with the noisy signals as the frequency sweeping may expand certain noise in the signal.
-> Thanks for the sharp point. As you said, frequency sweeping has the potential to expand certain noise in the signal. For example, when frequency sweeping is performed, signals are processed in areas other than the center frequency as well. Then, the area where the signal and noise are mixed is expanded, so a lot of noise can be mixed compared to the existing center frequency. As the next study, we are preparing a study to improve the proposed method so that robustness can be maintained even when signals are crossed between heterogeneous signals. In terms of frequency sweeping, heterogeneous signals such as OOK and FSK are noise, so we plan to add a preprocessor that can remove them. In the course of this, I will clarify by validating the possible performance when dealing with noisy signals as pointed out by you.
- For some piezo-electric transducers, there will be multiple resonance peaks in a wide range of frequencies, specifically at the harmonic frequency of the “center frequency”. Thus, the transmission curve may include multiple peaks, and a careful bandwidth selection should be made to define the range of f-20Db.
-> That’s right. Since we are targeting a typical, easy-to-use narrowband transducer, there is only one resonant peak at the fundamental frequency. However, in the case of a wide band transducer, harmonic frequencies may be included, so the frequency sweep range delta f must be carefully selected so that harmonic frequencies are not included. A note was added in 3.2.
Reviewer 3 Report
The article "Frequency sweep of CDMA encoding to reduce ultrasonic crosstalk" is interesting as well as original in some concepts attached here the observations that I understand must be resolved: 1.- I consider that the paragraph where the organization of the article is mentioned should be eliminated lines 84 to 90, they do not contribute to the presentation of the article. 2.- what happens when what is stated in lines 126 and 127 does not happen, that is, it is not 0 or 1, that is, how does this affect the system if it fails? 3.- If the frequency sweep range, ∆?, is set too wide, the ratio of the time of the area where the signal size to the time of the one-bit signal is small increases, which is not, why??? (Lines 160 and 161) 4.- Explain why they used the components in the ultrasonic transceiver system block diagram for proposed verification algorithm. 5.- Mention what are the limitations of the non-external system, of the system itself specifically. 6.- how to measure or understand the robust word of your system (line 364) 7.- In the experimentation they did not carry out tests modifying the angle between sensors and distances in an area of 100 square meters, why?
Author Response
1.- I consider that the paragraph where the organization of the article is mentioned should be eliminated lines 84 to 90, they do not contribute to the presentation of the article.
We eliminate it.
2.- what happens when what is stated in lines 126 and 127 does not happen, that is, it is not 0 or 1, that is, how does this affect the system if it fails?
This has been added to the conclusion.
This is the answer to the effect on the system if a value other than 0 or 1 of the frequency sweep method is received during reception, i.e. a heterogeneous signal OOK, FSK or PSK is received. The demodulator proposed in this paper sets the value obtained by multiplying the correlation value of the X mask by a coefficient as a threshold value. It is determined whether 1 is received, 0 is received, or simultaneously received according to whether each correlation value of 1 mask and X mask, and 0 mask and X mask is greater or less than the threshold. Even if an arbitrary signal is received, if the correlation value of the 1 and 0 mask is less than the threshold, code 0 or 1 is treated as not being received. OOK and PSK is detected differently depending on the threshold coefficient value because they have most of the energy at the center frequency. If the threshold coefficient is increased, OOK and PSK are not detected, but the probability that the method proposed in this paper will not be detected also increases. In the case of FSK, since energy is located at a frequency shifted from the center frequency, it is hardly detected.
3.- If the frequency sweep range, ∆?, is set too wide, the ratio of the time of the area where the signal size to the time of the one-bit signal is small increases, which is not, why??? (Lines 160 and 161)
Sorry, our mistake. “if the frequency sweep range, , is set too wide, the ratio of the time of the area where the signal size to the time of the one-bit signal is small increases, which is not good.” Is right sentence.
Owing to the bandwidth limitation of the ultrasonic transducer, the intensity of the signal deviating a lot from the center frequency band is very small. Therefore, if ∆f becomes too large, a signal is transmitted only in the center frequency band and no signal is transmitted/received in other frequency bands. Therefore, a signal is transmitted only for a small fraction of 1 bit signal time. So this is not good.
4.- Explain why they used the components in the ultrasonic transceiver system block diagram for proposed verification algorithm.
The previous explanation was sparse, so the first paragraph of 4.1 was revised to the following.
The ultrasonic transceiver system consists of personal computer (PC), microcontroller unit (MCU), amplifier (AMP), driver, and ultrasonic transducer. The ultrasonic transducer has one receiver (Rx) and several transmitters (Tx) to configure the environment where crosstalk occurs. The driver amplifies the power of the MCU's transmit signal to increase the strength of the ultrasonic transmit signal. The amplifier amplifies the voltage of the signal of the ultrasonic receiver and transmits it to the analog digital converter (ADC) of MCU. The signal-to-noise ratio (SNR) of the signal is improved by using a low noise amplifier. The MCU receives the trigger command from the PC, generates a transmit signal, and transfers the received signal to the PC. It can control the frequency in a burst form so that the transmission signal can be modulated with the signal of the proposed method and the existing method. It can also adjust the time delay so that the point of crosstalk can be changed. The PC demodulates the received signal and executes the CDMA algorithm.
5.- Mention what are the limitations of the non-external system, of the system itself specifically.
This has been added to the conclusion.
There are some limitations. First, in the case of a moving object, the Doppler effect, that is, a frequency shift phenomenon, occurs due to the speed difference. The greater the speed difference, the greater the frequency shift. The proposed demodulator performs code detection by correlating the code mask based on the center frequency. If a slight frequency shift occurs, demodulation will be performed normally. When a large frequency shift occurs, the correlation process is not normally performed because the spectrogram of the received signal deviates a lot from the center frequency. Therefore, there is a problem that the code cannot be detected. Second, when a heterogeneous signal other than the signal of the proposed method is received, a demodulation error may occur. The demodulator proposed in this paper sets the value obtained by multiplying the correlation value of the X mask by a coefficient as a threshold value. It is determined whether 1 is received, 0 is received, or simultaneously received according to whether each correlation value of 1 mask and X mask, and 0 mask and X mask is greater or less than the threshold. Even if an arbitrary signal is received, if the correlation value of the 1 mask and 0 mask is less than the threshold, code 0 or 1 is treated as not being received. OOK and PSK are detected differently depending on the threshold coefficient value because they have most of the energy at the center frequency. If the threshold coefficient is increased, OOK and PSK are not detected, but the probability that the method proposed in this paper will not be detected also increases. In the case of FSK, since energy is located at a frequency shifted from the center frequency, it is hardly detected. Third, multi-path fading effect is not considered. Since the reflected signal is also detected by the code, it is delayed as much as the fading TOF result, which may cause an error. In order to reduce the effect on multi path fading, there should be sufficient time interval between codes.
6.- how to measure or understand the robust word of your system (line 364)
Robust in our system means correct code demodulation in signal crosstalk situation. When TOF is measured through CDMA after code demodulation, if a code demodulation error occurs, the TOF error becomes large. Signal crosstalk is a crosstalk situation in which demodulation is difficult when the code length is shortened and the overlap ratio is increased. Therefore, to measure robustness, TOF accuracy according to code length was measured, and code detection rates according to overlap ratio were compared. As a result of the experiment, the proposed method has higher TOF accuracy even when the code length is shorter than that of the existing method, so it can be said that it is more robust to crosstalk. In addition, since the detection rate is maintained high even when the degree of overlap is high, it can be said that it is more robust to crosstalk situations.
7.- In the experimentation they did not carry out tests modifying the angle between sensors and distances in an area of 100 square meters, why?
In the case of the experiment to change the angle, the experiment was not conducted because it was judged that it was not suitable to verify the suggestion of this paper. The angular characteristics of the ultrasonic sensor were placed in a flat range, and this paper did not need to consider the effect of angle as an algorithm to show robustness to crosstalk. Because the phase of the signal does not change or the intensity does not change in crosstalk even if the angle is changed. The experiment of changing the distance was performed to vary the signal strength in the 1-bit overlap experiment. The experiment was carried out while changing the distance to 1.89m, 2.53m, 3.83m, and 7.70m where the signal strength was 0.8, 0.6, 0.4, and 0.2, respectively, based on 1.5m.

Round 2
Reviewer 1 Report
Thanks for the revised version and explanations.
Reviewer 3 Report
Cheers I do not have any observation of the work presented and corrected Thank you